# HR-pQCT and 3D Printing for Forensic and Orthopaedic Analysis of Gunshot-Induced Bone Damage

**DOI:** 10.3390/biomedicines13071742

**Published:** 2025-07-16

**Authors:** Richard Andreas Lindtner, Lukas Kampik, Werner Schmölz, Mateus Enzenberg, David Putzer, Rohit Arora, Bettina Zelger, Claudia Wöss, Gerald Degenhart, Christian Kremser, Michaela Lackner, Anton Kasper Pallua, Michael Schirmer, Johannes Dominikus Pallua

**Affiliations:** 1Department of Orthopaedics and Traumatology, Medical University of Innsbruck, Anichstraße 35, 6020 Innsbruck, Austria; richard.lindtner@i-med.ac.at (R.A.L.); lukas.kampik@i-med.ac.at (L.K.); werner.schmoelz@i-med.ac.at (W.S.); mateus.enzenberg@i-med.ac.at (M.E.); david.putzer@i-med.ac.at (D.P.); rohit.arora@tirol-kliniken.at (R.A.); 2Institute of Pathology, Neuropathology, and Molecular Pathology, Medical University of Innsbruck, Muellerstrasse 44, 6020 Innsbruck, Austria; bettina.zelger@i-med.ac.at; 3Institute of Forensic Medicine, Medical University of Innsbruck, Muellerstraße 44, 6020 Innsbruck, Austria; claudia.woess@i-med.ac.at; 4Core Facility MicroCT, University Clinic for Radiology, Medical University of Innsbruck, Anichstraße 35, 6020 Innsbruck, Austria; gerald.degenhart@i-med.ac.at; 5Department of Radiology, Medical University of Innsbruck, Anichstraße 35, 6020 Innsbruck, Austria; 6Institute for Hygiene and Medical Microbiology, Medical University of Innsbruck, 6020 Innsbruck, Austria; michaela.lackner@i-med.ac.at; 7Former Institute for Computed Tomography-Neuro CT, Medical University of Innsbruck, Anichstraße 35, 6020 Innsbruck, Austria; anton.k.pallua@cnh.at; 8Specialist Practise for Internal Medicine/Rheumatology, 6060 Hall in Tyrol, Austria; schirmer.michael@icloud.com

**Keywords:** 3D printing, HR-pQCT, micro-CT, gunshot trauma, osteomyelitis, forensic reconstruction, orthopaedic surgery, additive manufacturing, bone pathology, multimodal imaging

## Abstract

**Background/Objectives:** Recent breakthroughs in three-dimensional (3D) printing and high-resolution imaging have opened up new possibilities in personalized medicine, surgical planning, and forensic reconstruction. This study breaks new ground by evaluating the integration of high-resolution peripheral quantitative computed tomography (HR-pQCT) with multimodal imaging and additive manufacturing to assess a chronic, infected gunshot injury in the knee joint of a red deer. This unique approach serves as a translational model for complex skeletal trauma. **Methods:** Multimodal imaging—including clinical CT, MRI, and HR-pQCT—was used to characterise the extent of osseous and soft tissue damage. Histopathological and molecular analyses were performed to confirm the infectious agent. HR-pQCT datasets were segmented and processed for 3D printing using PolyJet, stereolithography (SLA), and fused deposition modelling (FDM). Printed models were quantitatively benchmarked through 3D surface deviation analysis. **Results:** Imaging revealed comminuted fractures, cortical and trabecular degradation, and soft tissue involvement, consistent with chronic osteomyelitis. *Sphingomonas* sp., a bacterium that forms biofilms, was identified as the pathogen. Among the printing methods, PolyJet and SLA demonstrated the highest anatomical accuracy, whereas FDM exhibited greater geometric deviation. **Conclusions:** HR-pQCT-guided 3D printing provides a powerful tool for the anatomical visualisation and quantitative assessment of complex bone pathology. This approach not only enhances diagnostic precision but also supports applications in surgical rehearsal and forensic analysis. It illustrates the potential of digital imaging and additive manufacturing to advance orthopaedic and trauma care, inspiring future research and applications in the field.

## 1. Introduction

The rapid advancement of digitalisation and additive manufacturing has transformed surgical planning, forensic investigations, and personalised medicine. Since the advent of Industry 4.0, data-driven applications have enhanced medical imaging and patient-specific interventions, while Industry 5.0, introduced in 2015, integrates artificial intelligence (AI) and human-centred approaches into healthcare innovations [1]. One of the most impactful developments in this field is three-dimensional (3D) printing, which rapidly fabricates complex anatomical structures, facilitating precise preoperative planning, surgical training, and personalised implants [2,3]. Initially developed for prototyping in industries such as aerospace and automotive, 3D printing has revolutionised biomedical applications, particularly in orthopaedic trauma and reconstructive surgery [3,4]. Three-dimensional printing is particularly valuable for managing complex fractures, joint injuries, and ballistic trauma in orthopaedics and traumatology. By combining advanced imaging techniques with computer-aided design (CAD), 3D printing enables the extracorporeal reproduction of intricate bone structures, offering both tactile and visual feedback beyond traditional two-dimensional (2D) imaging modalities, such as X-ray, computed tomography (CT), and magnetic resonance imaging (MRI) [4,5,6]. These detailed models enhance surgical precision and improve patient outcomes, especially in cases requiring preoperative simulation and implant customisation [6,7]. Gunshot injuries pose unique challenges in orthopaedic and forensic medicine due to high-energy impact forces, extensive bone fragmentation, and infection risks. In ballistic trauma, bone undergoes cavitation effects and secondary fractures, leading to nonunion of the fracture and osteomyelitis [6,8]. Integrating high-resolution imaging and 3D printing provides a powerful tool for visualising injury patterns, aiding clinical treatment strategies and forensic investigations. A breakthrough in bone imaging has been the development of high-resolution peripheral quantitative computed tomography (HR-pQCT). Compared to dual-energy X-ray absorptiometry (DXA), which assesses bone mineral density (BMD) without structural details, HR-pQCT enables three-dimensional visualisation of trabecular and cortical bone microarchitecture [9,10,11,12]. First introduced in 2005 by Scanco Medical AG, HR-pQCT has since evolved into the XtremeCT II scanner, offering an isotropic voxel size of 61 µm at low radiation exposure (5 µSv per scanned cm) [11,12]. This technology allows for detailed assessments of osteoporosis, fracture healing, and post-traumatic bone remodelling, making it invaluable in orthopaedic trauma research [13,14,15]. Compared to micro-CT, which provides higher spatial resolution but is limited to ex vivo applications, HR-pQCT enables non-invasive, in vivo imaging with high diagnostic accuracy at the extremities [16]. Recent applications of HR-pQCT include analysing inflammatory joint disorders, implant stability, and the effects of pharmacological treatments, demonstrating its expanding role in orthopaedic and forensic medicine [17,18,19,20]. This study hypothesises that anatomical models generated from HR-pQCT data can be accurately reproduced using additive manufacturing techniques and that different 3D printing technologies will exhibit significant variability in dimensional fidelity. This work aims to benchmark and compare the performance of several commonly used 3D printing technologies by analysing their ability to replicate a gunshot-injured bone using HR-pQCT data. We also explore how these models might serve as educational and forensic tools in reconstructive and trauma surgery. This study employed HR-pQCT imaging and additive manufacturing to investigate an infected gunshot injury in the knee of a red deer (*Cervus elaphus*). The primary objective was to evaluate microstructural bone integrity, characterise pathological changes, and assess the accuracy of 3D-printed models generated from micro-CT data using various additive manufacturing technologies. We systematically analysed the fidelity of PolyJet (J35 (Stratasys Ltd., Rehovot Israel)), Stereolithography (SLA, envisionOne (EnvisionTEC GmbH, Gladbeck, Germany)), and Fused Deposition Modelling (FDM, Fortus 450MC (Stratasys Ltd., Rehovot, Isreal), Raise3D E2 (Raise 3D Technologies Inc., Irvine, CA, USA), Prusa XL (Prusa Research, Prague, Czech Republic), Bambu Lab A1 Mini (Bambu Lab, Shenzhen, China), WASP (CSP srl—WASP, Massa Lombarda, Italy)) to determine their accuracy, reproducibility, and suitability for forensic and surgical applications. By systematically benchmarking 3D printing techniques against HR-pQCT data and aligning digital imaging and physical modelling, this study offers valuable insights into the reliability and clinical applicability of 3D printing technologies in orthopaedic trauma, reconstructive surgery, and forensic investigations. These findings contribute to the optimisation of digital imaging and additive manufacturing, enabling improved preoperative planning, injury documentation, and treatment strategies in complex trauma cases.

## 2. Materials and Methods

### 2.1. Study Design

This was a single-case, proof-of-concept study utilising a micro-CT dataset of an injured knee joint of a red deer to evaluate the dimensional accuracy and fidelity of 3D-printed anatomical models. While limited in sample size, this approach allowed for detailed benchmarking across different additive manufacturing technologies. Due to the forensic nature and anatomical rarity of the specimen, additional specimens were not feasible at this stage. All models underwent post-processing according to manufacturer guidelines. To assess dimensional accuracy and geometric fidelity, each 3D-printed model was rescanned using HR-pQCT under identical imaging conditions to those for the original bone specimen. The reconstructed scans were then compared with the original micro-CT dataset through surface congruency analysis, wall thickness deviation mapping and quantitative dimensional error assessment using 3D engineering software (Analyze 14.0, Mayo Foundation, Rochester, MN, USA)

By employing multiple 3D printing technologies and benchmarking their performance against high-resolution imaging, this study provides insights into the accuracy, reproducibility, and feasibility of different 3D printing methods for applications in orthopaedic research, forensic science, and surgical planning.

### 2.2. Autopsy

A cadaveric limb of a red deer (*Cervus elaphus*) with a documented gunshot wound was obtained for gross examination and dissection. The specimen was obtained through a collaboration with professional hunters from the Österreichische Bundesforste and used under institutional biosafety regulations. As the study used a non-human specimen and did not involve live animal experimentation or human tissue, formal ethics committee approval was not required. We confirm that all procedures were conducted following institutional and legal standards for post-mortem tissue handling. The specimen was preserved at 4 °C and handled by standard biosafety protocols to prevent contamination. The limb was positioned supine on a sterile dissection table under controlled lighting conditions. An initial external inspection was performed to document the wound entry characteristics, surrounding tissue condition, and presence of any exudates. Photographic documentation was conducted using a high-resolution digital camera (Nikon D800E, Nikon Corporation, Tokyo, Japan) with standardised lighting to ensure consistent image capture. A systematic stepwise dissection was performed using sterile scalpels and forceps, following standard forensic and anatomical dissection protocols. The skin was incised along the wound trajectory, and underlying tissues were carefully separated to expose deeper structures. Haemorrhagic regions, necrotic tissue, and any purulent material were visually assessed and recorded at each stage of dissection. A sequential examination of tissue layers, including subcutaneous tissue, fascia, muscle, and bone, was conducted to assess the extent of injury and signs of infection. Gross examination focused on identifying tissue damage, haemorrhagic infiltration, necrotic changes, and disease. The following parameters were documented:Skin and subcutaneous tissue: Fibrin deposits, inflammatory exudate, necrosis, and tissue discolouration.Fascial layers: Structural integrity, tissue separation, and signs of enzymatic degradation.Muscular structures: Degree of muscle fibre disruption, haemorrhaging, and evidence of cavitation effects.Bone examination: Cross-sectional cortical and trabecular bone analysis for fractures, marrow involvement, and early osteomyelitis indicators.

### 2.3. Computed Tomography Imaging

CT images were acquired using a clinical CT scanner (SOMATOM Confidence, Siemens Healthineers, Erlangen, Germany). The knee joint of a red deer was scanned with the following parameters: tube voltage of 120 kVp, maximum X-ray tube current of 184 mA, X-ray tube current of 112 mA, exposure time per rotation of 1 s, a nominal single collimation width of 0.6 mm, nominal total collimation width of 12 mm, and a pitch factor of 0.55.

### 2.4. HR-pQCT Scan Acquisition

Scans were obtained using a second-generation HR-pQCT system (XtremeCT II, Scanco Medical, Brüttisellen, Switzerland). The HR-pQCT utilises the manufacturer-provided software suite µCT v. 6.1. The scans were visualised with three stacks of 10.2 mm. Pre-settings of all scans included a resolution of 60.7 µm isovoxels, resulting in 504 slices, an integration time of 46 ms per projection, 900 projections, and a voltage and intensity of 68 kV and 1460 µA, respectively. According to the manufacturer’s settings, the radiation dose per stack was 5 µSv, so the maximum radiation dose was 18 µSv. For reconstruction, the manufacturer utilised a filtered backprojection algorithm.

### 2.5. Magnetic Resonance Imaging (MRI) Data Acquisition and Processing

MRI was performed using a 3-Tesla Scanner (Magnetom Skyra, Siemens Healthineers, Erlangen, Germany) using an 18-channel phased-array body coil together with the integrated 32-channel spine matrix coil. A 3D T1-weighted spoiled gradient echo sequence (FLASH) was used [21,22], with the following parameters: TR = 11 ms, TE = 4.76 ms, flip angle: 20°, receive bandwidth: 180 Hz/pixel, FOV = 250 mm, image matrix = 320 × 320, voxel size = 0.78 mm × 0.78 mm × 0.8 mm. In addition, a T2-weighted turbo spin-echo sequence with short TI inversion recovery (STIR) fat suppression was acquired with TR = 4900 ms, TE = 61 ms, TI = 200 ms, echo train length = 12, slice thickness = 3 mm, spacing between slices = 3.6 mm, acquisition matrix = 384 × 312, FOV = 250 mm, voxel size = 0.65 mm × 0.65 mm × 3 mm, number of slices: 25. Data processing and analyses were performed using Syngo.Via (Siemens Healthineers, Erlangen, Germany).

### 2.6. Three-Dimensional Printing Methods

The injured knee joint micro-CT dataset was processed and converted into an STL (stereolithography) format to evaluate the dimensional accuracy and fidelity of the 3D-printed models. The digital model was fabricated using multiple 3D printing technologies, each representing a distinct additive manufacturing approach (see Table 1).

After fabrication, all 3D-printed models underwent post-processing following manufacturers’ guidelines. Each model was then rescanned using micro-CT under the same imaging conditions as the original bone specimen to assess their dimensional accuracy and geometric fidelity. The reconstructed scans were then compared with the original micro-CT dataset through:Surface congruency analysis;Wall thickness deviation mapping;Quantitative dimensional error assessment using 3D engineering software (Analyze 14.0, Mayo Foundation, USA).

By employing multiple 3D printing technologies and benchmarking their performance against high-resolution imaging, this study provides insights into the accuracy, reproducibility, and feasibility of different 3D printing methods for applications in orthopaedic research, forensic science, and surgical planning.

### 2.7. Application of Quantification Methods for Image Analyses

Image analyses were performed using the HR-pQCT dataset from the red deer knee joint. Reconstructions were completed using Analyze 14.0 (Analyze Direct Inc., Overland Park, KS, USA) and GOM Inspect Version 2025 (GOM, Braunschweig, Germany) software. In the first stage of assessment, the CT scan of the knee joint was reconstructed via semi-automatic object segmentation with region grow (minimum 319, maximum 3071).

### 2.8. Three-Dimensional (3D) Measurement Method

For 3D measurements based on HR-pQCT data, all scans were reconstructed in 3D space using 3D Slicer 5.6.1 (http://www.slicer.org). This software enabled the conversion of a DICOM file into a Mesh file, which was further evaluated using the GOM Inspect Version 2025 (GOM, Braunschweig, Germany) software. One observer conducted 3D reconstruction to avoid any errors associated with the 3D reconstruction itself. The CT scans were designated as “model” and converted into the CAD format, while the second tomogram, referred to as the “comparative”, was converted into the Mesh format, a universal file format for geometry. Both models were compared using GOM Inspect (GOM, Braunschweig, Germany). Next, all reconstructed knee joint models were aligned over the red-marked area. Segmentation thresholds in 3D Slicer were set between 319 and 3071 HU to isolate trabecular and cortical bone while excluding soft tissue. The STL models retained the internal trabecular and medullary architecture without infilling, thereby preserving anatomical fidelity. STL file sizes ranged from 50 to 120 MB and contained 2.1 to 4.8 million triangles, depending on the printer resolution and slicing strategy.

### 2.9. Histological and Molecular Analysis

Before histological and molecular analysis, specimens were fixed in formalin and embedded in paraffin. Paraffin-embedded blocks were mounted on a microtome for tissue sectioning, and 3.0 µm-thick sections were obtained. These sections were stained with Periodic Acid-Schiff (PAS), Chromotrope Aniline Blue Special Stain (CAb), Grocott’s Methenamine Silver (GMS), and Hematoxylin and Eosin (H&E) to facilitate histopathological assessment. For digital analysis, slides were scanned using a Pannoramic SCAN digital slide scanner (3DHISTECH, Budapest, Hungary) equipped with a plan-apochromatic objective (20× magnification, numerical aperture: 0.8). Histological evaluation and infection scoring were performed using Pannoramic Viewer software ver. 2.3 (3DHISTECH, Budapest, Hungary). An experienced pathologist meticulously reviewed and characterised all tissue sections for signs of infection. For DNA extraction, four 5 µm-thick FFPE (formalin-fixed, paraffin-embedded) tissue sections per sample were processed using the BioRobot EZ1 (Qiagen, Hilden, Germany) and the EZ1 DNA Tissue Kit (Qiagen) following the manufacturer’s protocol (Purification of DNA from Paraffin-Embedded Tissue). The elution volume was set to 50 µL. Samples were subsequently sequenced for Seq GP16 and Seq GN16.

### 2.10. Statistical Analysis

Statistical analyses were performed using GraphPad Prism version 10 (GraphPad Prism Software, Inc. version 9, San Diego, CA, USA) and IBM SPSS Statistics version 22.0 (IBM Corp., Armonk, NY, USA). After testing for normal distribution, linear and volumetric measurements were compared using one-way ANOVA and Fisher’s LSD post hoc test. A *p*-value < 0.05 was considered statistically significant.

## 3. Results

### 3.1. Gross Examination and Dissection Findings

Figure 1A shows the initial dissection of the gunshot-injured limb, which revealed severe tissue damage. The entry wound was partially obscured by inflammatory exudate and fibrin deposits, while irregular tearing and contusions in the surrounding skin indicated a high-velocity projectile impact. The presence of pus suggested a secondary bacterial wound infection. The edges of the skin appeared necrotic. The findings are consistent with a high kinetic energy transfer from a bullet. The ammunition or calibre used is unknown, and the period between the gunshot wound and death is also unknown. Possibly, cavitation effects occurred because bone is involved, and the injury may have resulted from a short-range shot or hunting ammunition, such as a partial jacket bullet. In Figure 1B, deeper layers of tissue were exposed, showing severe muscular destruction along the bullet path.

The muscle fibres appeared disrupted and macerated, with extensive necrosis. The loss of clear tissue structure suggested high-energy impact trauma, possibly associated with cavitation effects that led to widespread tissue liquefaction. Visible pockets of pus indicated secondary bacterial infection in the wound tract. The discolouration and structural breakdown suggested a chronic infection, possibly involving necrotising fasciitis. In Figure 1C, the gunshot trajectory was further delineated, with significant muscle loss and deep-seated infection. The presence of pus in the deeper tissue layers suggested an advanced state of infection, possibly involving anaerobic bacteria. The soft tissues exhibited a combination of coagulative necrosis, due to ischemia caused by the bullet’s destruction of blood vessels, and liquefactive necrosis, resulting from infection. The fascial layers were partially separated, and the tissue appeared fragile and detached, consistent with progressive necrosis and enzymatic degradation of infected tissue. In Figure 1D, the bone was cross-sectioned, revealing damage consistent with a projectile impact. The cortical bone showed signs of fracturing but no extensive bone marrow involvement. The absence of purulent material within the bone cavity suggested that, although soft tissues appeared severely infected, the osseous structures had not yet progressed to advanced osteomyelitis. However, focal bone erosion and discolouration indicated an early periosteal reaction and potential osteonecrosis. Overall, gross examination revealed a severe gunshot-related infection with extensive soft tissue necrosis and localised bone damage. While the infection appeared primarily confined to the soft tissues, signs of early bone involvement warranted close monitoring.

### 3.2. Morphological Analysis via CT, HR-pQCT and MRI

To comprehensively evaluate the extent of bone and soft tissue damage resulting from high-energy ballistic trauma, a multimodal imaging approach combining CT, MRI, and HR-pQCT was employed. These complementary modalities enabled detailed evaluation of structural disruption, soft tissue pathology, and infection progression.

The CT scan (Figure 2A) revealed extensive morphological alterations of the affected limb, including pronounced cortical destruction, fragmentation, and periosteal reaction. The fracture pattern was markedly comminuted, with irregular and jagged bone margins, consistent with high-velocity impact injury. Trabecular discontinuities and cortical gaps were evident, and the disrupted bone matrix suggested mechanical instability and impaired healing. In the surrounding soft tissues, areas of heterogeneous radiodensity suggested the presence of hematoma, necrosis, or inflammatory infiltration. A periosteal reaction with irregular new bone formation further indicated a chronic inflammatory response, raising suspicion for secondary osteomyelitis.

MRI (Figure 2B) provided enhanced soft tissue contrast and sensitivity to bone marrow pathology. The trajectory of the projectile was visualised, accompanied by extensive oedema and signal alterations along the wound tract. Hyperintense regions on T1-weighted images indicated fluid accumulation within the musculature and surrounding soft tissue, likely representing a combination of haemorrhage, interstitial oedema, and potential abscess formation. Hypointense areas were suggestive of tissue necrosis or fibrosis. Abnormal signal intensities within the bone marrow were consistent with early changes in osteomyelitis. In conjunction with irregular wound margins and soft tissue displacement, these findings supported the presence of cavitation effects caused by energy transfer from the projectile. Such effects likely contributed to further tissue disruption, impaired healing, and increased risk of infection.

HR-pQCT (Figure 2C) allowed for a detailed assessment of the trabecular and cortical microarchitecture at the lesion site. The scans revealed profound trabecular loss, structural collapse, and cortical erosion, with evident focal bone resorption. The periosteal surface appeared irregular and fragmented, and areas of bone sequestration were identified, indicative of early-stage osteolysis. These fine structural alterations provided microanatomical confirmation of the gross damage seen on clinical CT and were strongly suggestive of bacterial infiltration and progressive inflammatory degradation, hallmark features of developing osteomyelitis.

Together, these imaging modalities provided a comprehensive overview of the complex pathophysiological processes following ballistic trauma, including mechanical destruction, inflammatory soft tissue involvement, and early-stage infection—all of which are essential considerations for clinical management and surgical decision-making.

### 3.3. Histological and Molecular Analysis of Infected Bone Tissue

Histological and molecular analyses were performed to investigate microbial involvement, bone degradation, and inflammatory responses in chronically infected bone and adjacent soft tissue. A panel of stains—Periodic Acid-Schiff (PAS), Chromotrope Aniline Blue Special Stain (CAb), Grocott’s Methenamine Silver (GMS), and Haematoxylin and Eosin (H&E)—was applied to serial sections of bone and soft tissue specimens.

In the preserved regions of trabecular bone (Figure 3A), PAS staining revealed intact lamellar architecture with fatty marrow and no signs of inflammatory cell infiltration. Calcofluor White staining (Figure 3B) showed no fungal elements, effectively ruling out fungal colonisation in these structurally intact zones.

In contrast, PAS staining of diseased bone (Figure 3C) demonstrated disrupted trabecular structure, with morphological features indicative of microbial infiltration. Grocott’s staining (Figure 3D) visualised biofilm-like bacterial colonies (“bacterial turf”) within porous and irregular trabeculae, without detection of fungal hyphae, suggesting a bacterial rather than fungal aetiology.

Fibrosclerotic remodelling within the bone was observed, as shown in PAS-stained sections (Figure 3E,F), with pronounced extracellular matrix deposition and fibroblast proliferation, consistent with chronic inflammation or post-infectious healing. PAS staining of actively infected bone (Figure 3G) revealed loss of trabecular integrity, accompanied by mononuclear infiltrates and signs of early necrosis. Fibrosclerotic remodelling refers to the pathological replacement of normal bone or soft tissue architecture with dense fibrous connective tissue and sclerotic (hardened) collagen matrices, typically resulting from chronic inflammation, infection, or impaired healing. This process reflects a late-stage tissue response characterised by reduced vascularity, limited regenerative capacity, and structural stiffening. H&E staining confirmed advanced inflammatory pathology in the most severely affected regions (Figure 3H), characterised by dense infiltrates of neutrophils and lymphocytes, osteolytic destruction, fat necrosis, and vascular congestion—hallmarks of bacterial osteomyelitis.

Molecular sequencing corroborated the histological findings by detecting *Sphingomonas* spp., a biofilm-forming bacterium associated with nosocomial infections. No fungal DNA was identified, reinforcing the bacterial origin of the infection.

### 3.4. HR-pQCT Imaging, 3D Volumetric Reconstruction, and Anatomical Dissection

HR-pQCT imaging, 3D volumetric reconstruction, and anatomical dissection were performed to assess the extent of osseous damage, fragmentation, and infection. The imaging modalities provided high-resolution insights into cortical and trabecular disruption, overall bone integrity, and the relationship between structural damage and infection progression. A 3D-printed model was also generated to compare anatomical findings with digital reconstructions for surgical planning and forensic analysis.

The axial HR-pQCT image (Figure 4A) revealed extensive bone fragmentation, characterised by multiple displaced and detached osseous fragments. The cortical bone appeared disrupted, with discontinuities along the fracture lines, indicative of a high-velocity projectile impact. The trabecular structure was irregular, with loss of continuity in several regions, suggesting bone comminution and mechanical instability. Additionally, radiolucent areas (darker regions) within the trabecular bone suggested osteolysis, likely caused by chronic infection or inflammatory bone resorption. Small osseous sequestra were visible, indicating devitalised bone fragments within the soft tissue, serving as a potential nidus for chronic osteomyelitis development. The heterogeneity of bone density further suggested regions of reactive bone formation, reflecting ongoing reparative processes in response to trauma and infection.

The coronal CT scan (Figure 4B) offers a frontal view, demonstrating severe vertical displacement of bone fragments. The cortical integrity was significantly compromised, with widened fracture gaps and uneven bony surfaces. There was evidence of trabecular disorganisation, which, in conjunction with periosteal irregularities, suggested a possible infectious or inflammatory process at the fracture margins. Areas of increased radiodensity adjacent to the fracture site may represent sclerotic bone formation, an adaptive response to chronic mechanical stress and prolonged inflammation. These findings suggested that the bone has undergone partial remodelling, but osteolytic regions raised concerns for persistent infection and nonunion healing.

The sagittal CT scan (Figure 4C) provides a longitudinal view of the injury, demonstrating a full-thickness cortical breach with extensive trabecular fragmentation. The depth of structural damage suggested that the projectile likely caused cavitation effects, leading to progressive internal bone disintegration. The cortical surface was highly irregular, with areas of erosion and discontinuity, which could indicate periosteal reaction or chronic osteomyelitis changes. In addition, the trabecular bone appeared highly porous, with multiple voids that could represent either infection-induced bone resorption or structural failure due to trauma. The adjacent soft tissue seemed to have indirect signs of swelling, consistent with chronic inflammation or residual haemorrhage from the initial injury.

The 3D volumetric reconstruction (Figure 4D) allowed a detailed visualisation of the overall fracture morphology and bone loss. The volumetric data confirmed extensive comminuted fractures, with multiple small fragments dispersed throughout the region of interest. Additionally, there is evidence of cortical surface irregularity, which suggests progressive osteolysis or sequestration of devitalised bone. The 3D reconstruction highlighted both high-density and low-density regions, distinguishing areas of structural integrity from those affected by infection or resorption. These findings validated the severity of the bone damage and the potential necessity for aggressive surgical debridement to remove necrotic tissue.

The gross dissection image (Figure 4E) presents a macroscopic view of the damaged bone. The black arrow marks a necrotic bone fragment, which appears discoloured, irregular, and embedded within inflamed soft tissue. The adjacent soft tissue showed signs of fibrosis, swelling, and purulent material, suggesting a chronic inflammatory response or ongoing infection. Fibrotic adhesions between the bone and surrounding tissue indicated delayed healing and possible chronic osteomyelitis. The exposed bone surface showed regions of cortical erosion, further supporting the radiological findings of progressive bone degradation and incomplete remodelling.

A 3D-printed model of the affected bone (Figure 4F) was created to validate the HR-pQCT findings and enhance preoperative planning. This model accurately replicated the cortical disruptions, fracture lines, and trabecular voids seen in imaging. A direct comparison with an extracted bone fragment demonstrated high concordance between the digital and physical findings, confirming the accuracy of CT-based reconstructions for forensic and surgical applications. The physical manipulation of the printed model provided an additional diagnostic and planning advantage, allowing for the simulation of surgical approaches, implant fitting, and resection margins. This approach is particularly beneficial for complex fracture patterns, enabling more precise preoperative decision-making.

### 3.5. Accuracy and Performance Evaluation of 3D Printing Technologies

A comprehensive surface deviation and statistical analysis were conducted to evaluate the dimensional accuracy, precision, and performance of different 3D printing technologies. The study involved high-resolution 3D scanning, deviation mapping, quantitative statistical analysis, and performance benchmarking, allowing a detailed assessment of the accuracy and reproducibility of each printer. The findings provide valuable insights into the systematic deviations observed among different additive manufacturing technologies and their implications for precision manufacturing and biomedical applications.

#### 3.5.1. Dimensional Accuracy and Surface Deviation Analysis

Surface deviation mapping was performed using 3D scanning techniques to assess the accuracy of printed models compared to their digital reference counterparts. The analysis evaluated surface topology, material distribution, and deviation intensity across multiple measurement points.

The colour-coded deviation maps (Figure 5) illustrate regional variations in dimensional accuracy, where red represents areas of excessive material deposition (positive deviation), blue indicates material deficits (negative deviation), and green denotes near-nominal accuracy. The deviation maps revealed significant variations in print fidelity across different printing technologies, suggesting that material properties, printing methodology, and post-processing effects significantly influence print accuracy.

Fused Deposition Modelling (FDM) printers exhibited the highest surface deviations, with pronounced under-extrusion and over-extrusion effects, leading to irregularities in print geometry. Stereolithography (SLA) and PolyJet printers demonstrated superior precision, with minimal deviation from the reference model, confirming their suitability for applications requiring high-dimensional accuracy.

The statistical deviation analysis (Figure 5A) further corroborated the findings from 3D deviation mapping, showing a higher variation in deviation magnitudes among FDM-based printers, whereas PolyJet and SLA-based systems maintained more uniform print accuracy.

#### 3.5.2. Statistical Analysis of Dimensional Deviations

A quantitative assessment of mean deviation, standard deviation, and extreme values was conducted for all tested 3D printers. The results are summarised in Table 2, which highlights printer-specific deviations, their dispersion, and outlier effects.

The envisionOne, Fortus 450MC and Bambulab A1 Mini exhibited the lowest mean surface deviations, indicating superior precision in maintaining geometric accuracy across printing models. The Fortus 450MC and J35 Stratasys demonstrated moderate deviations, with their performance aligning with industrial-grade applications requiring controlled precision. In contrast, the PrusaXL and Raise3D printers showed substantially higher mean and standard deviations, reflecting both larger systematic errors and greater variability between prints. The Delta WASP system, while also based on FDM technology, exhibited a relatively high standard deviation and a significant spread in minimum and maximum values, likely attributable to the inherent variability in extrusion control characteristic of delta-based architectures. These findings underscore the influence of printing technology and machine design on dimensional accuracy and repeatability.

#### 3.5.3. Benchmarking of 3D Printing Technologies

To further evaluate the efficiency, material utilisation, and process feasibility of each 3D printing technology, printing time, material consumption, and post-processing requirements were analysed. The findings are summarised in Table 3, detailing technical specifications and processing conditions for each printing technology.

The PolyJet-based J35 Stratasys and SLA-based envisionOne achieved the highest precision with the finest layer thicknesses (18.75 µm and 50 µm, respectively), making them suitable for high-detail applications. FDM-based systems demonstrated larger layer thicknesses (200–254 µm), with more significant print time variability depending on model complexity and support structures.

Post-processing time varied significantly, particularly for RAISE3D E2, which required 96 h for water-soluble support removal. Sterilisation compatibility was found primarily in industrial-grade systems (J35 Stratasys and Fortus 450MC), while desktop FDM and SLA printers lacked specified sterilisation suitability.

#### 3.5.4. Three-Dimensional Deviation Mapping for Dimensional Accuracy Assessment

To assess dimensional fidelity across different 3D printing technologies, 3D surface deviation mapping was performed, providing a spatially resolved visualisation of deviations between printed models and their digital reference geometry. The false-colour maps shown in Figure 6 highlight localised expansion or contraction, with red regions indicating positive deviations (material excess), blue regions indicating negative deviations (material loss or shrinkage), and green representing near-nominal accuracy.

As illustrated in the grayscale reference model (top left), the original CAD structure served as the benchmark for deviation computation. The subsequent false-colour renderings capture deviations from multiple viewing angles, allowing for a comprehensive assessment of shape fidelity and manufacturing precision. Printers demonstrating higher accuracy—such as the SLA and PolyJet-based systems—exhibit uniform colour distributions concentrated in the green-to-yellow range, reflecting minimal geometric error. In contrast, FDM-based systems displayed more pronounced red and blue zones, indicative of systematic under-extrusion, over-extrusion, or warping.

The observed patterns closely align with the quantitative deviation metrics presented in Table 2 and print benchmarking outcomes summarised in Table 3. Printers such as the Bambulab A1 Mini and J35 Stratasys produced more geometrically consistent models, supporting their suitability for high-dimensional precision applications. Conversely, systems like PrusaXL and Raise3D showed significant variability, especially in critical structural zones, suggesting the need for process tuning and potential post-processing to meet strict tolerance thresholds.

These findings underscore the value of visual deviation mapping as a diagnostic tool in quality assurance and printer selection. The ability to spatially localise inaccuracies enables targeted process refinement and material optimisation, particularly in biomedical and engineering applications demanding high-resolution accuracy.

## 4. Discussion

Integrating high-resolution imaging modalities with additive manufacturing represents a paradigm shift in biomedical engineering, particularly in complex trauma, surgical planning, and forensic reconstruction. This study demonstrates the power of combining HR-pQCT, multimodal radiologic assessment, histopathological and molecular diagnostics, and 3D printing technologies to examine and reconstruct a chronically infected gunshot injury in a red deer knee joint—a translational model for human orthopaedic trauma. This study was designed as a single-case, proof-of-concept investigation using HR-pQCT data from a rare forensic specimen—an injured knee joint of a red deer. Although limited by its sample size, the unique anatomical features and chronic ballistic trauma of the specimen provided an exceptional opportunity for detailed benchmarking of additive manufacturing technologies. Due to the specimen’s forensic relevance and anatomical uniqueness, additional samples were not available at this stage.

Nevertheless, the case enabled a high-resolution, multimodal assessment of dimensional accuracy and model fidelity across different printing approaches. Multimodal imaging enabled a nuanced understanding of the pathophysiological consequences of ballistic trauma. CT and HR-pQCT imaging revealed extensive comminution, cortical fragmentation, and trabecular collapse—features characteristic of high-velocity bullet injuries. HR-pQCT provided high-resolution insight into early osteolytic changes and bone sequestration, central to developing post-traumatic osteomyelitis. MRI added critical value by highlighting soft tissue involvement, fluid accumulation, and abnormal bone marrow signals consistent with inflammatory infiltration, abscess formation, and chronic infection. This comprehensive radiologic assessment aligns with previous findings on the value of multimodal imaging for complex post-traumatic scenarios [6,17,19].

Histopathological and molecular analyses confirmed the presence of chronic bacterial osteomyelitis, with no evidence of fungal colonisation. Identification of *Sphingomonas* sp. supports the hypothesis that ballistic wounds, particularly when associated with retained necrotic tissue, serve as fertile ground for persistent infections that may evade standard antibiotic therapy. The granulomatous remodelling, fat necrosis, and inflammatory infiltrates further underscore these injuries’ chronicity and clinical complexity.

A key innovation of this study is using HR-pQCT-derived datasets to generate anatomically precise 3D-printed models. These models served as tangible, manipulable reconstructions of the injured site, enabling improved visualisation of the fracture morphology and infection zones. Quantitative surface deviation mapping revealed that PolyJet and SLA printers outperformed FDM-based systems in replicating bone geometry with minimal deviation. These findings are consistent with comparative studies on additive manufacturing fidelity in biomedical contexts [3,7]. Although direct caliper-based dimensional validation against the original bone was not feasible due to structural complexity, all printed models were rescanned using high-resolution HR-pQCT under standardised conditions. A subsequent 3D surface deviation analysis against the original micro-CT data enabled a robust and reproducible assessment of geometric fidelity, ensuring submillimetric accuracy despite the absence of manual measurements.

From a translational standpoint, the high-dimensional accuracy of resin-based printing methods underscores their suitability for applications requiring fine anatomical detail, such as preoperative simulation, surgical guide fabrication, and forensic documentation. The type of polymer used in additive manufacturing has a substantial impact on the dimensional accuracy, surface resolution, and clinical applicability of printed models. In this study, most FDM printers—including the Raise3D E2 and Prusa XL—utilised polylactic acid (PLA), a biodegradable thermoplastic known for ease of printing and cost-efficiency. The Prusa XL additionally employed BVOH as a water-soluble support material. While PLA provides sufficient geometric stability for educational and planning purposes, its mechanical limitations and low heat resistance restrict its use in load-bearing or sterilisable clinical applications.

In contrast, the Fortus 450 used acrylonitrile butadiene styrene (ABS), a more robust and sterilisable material, suitable for industrial and surgical workflows. PolyJet and SLA systems employed proprietary photopolymers with high resolution and surface fidelity, though with more complex post-processing requirements and higher material costs. These differences underline the importance of matching material properties to intended applications in biomedical modelling. Conversely, FDM printers, despite lower precision, remain valuable for educational tools, cost-sensitive prototyping, and preliminary assessments, particularly when high geometric accuracy is not essential.

This integrative approach, combining digital imaging, molecular diagnostics, and physical model fabrication, exemplifies the direction of modern personalised medicine. In orthopaedics, such workflows allow for patient-specific planning, custom implant design, and simulation of surgical procedures. In forensic medicine, 3D-printed reconstructions derived from validated imaging offer court-admissible evidence, enabling visualisation of trauma patterns, projectile trajectories, and injury mechanisms.

Nevertheless, certain limitations should be acknowledged. Material differences between printed polymers and biological bone restrict the use of printed models for mechanical testing or load-bearing simulations unless specifically tailored materials are employed. This study focused primarily on the geometric fidelity of anatomical models rather than their mechanical performance. As such, mechanical testing of the printed materials was not performed. However, it is essential to note that the mechanical behaviour of polymers such as PLA or photopolymer resins differs significantly from native bone, and thus these models are not suitable for load-bearing simulations without further validation. Moreover, potential effects such as stress shielding—where differences in elastic modulus between implant and host tissue alter physiological loading patterns—were not evaluated in this proof-of-concept design. Future studies should consider incorporating finite element analysis and standardised mechanical testing protocols to assess biomechanical behaviour and explore the role of material stiffness and model design in stress distribution and surgical rehearsal. Furthermore, additive manufacturing remains time- and resource-intensive, posing challenges to widespread clinical implementation. Future research should explore multi-material printing, the integration of bioresorbable materials, and bioprinting technologies to bridge the gap between digital reconstruction and clinical application.

In conclusion, this study highlights the synergistic value of HR-pQCT imaging and 3D printing in analysing and reconstructing complex bone trauma. By systematically benchmarking additive manufacturing technologies and validating radiologic findings with histological and molecular methods, we provide a robust framework for translational applications in trauma surgery, personalised medicine, and forensic science. These findings contribute directly to the core objectives of this special issue, demonstrating how digital-to-physical conversion technologies can enhance diagnostic precision, surgical planning, and biomedical innovation.

## 5. Conclusions

This study demonstrates the integrative potential of HR-pQCT, multimodal imaging, and additive manufacturing in the forensic and orthopaedic analysis of gunshot-induced bone trauma. By combining radiologic, histopathologic, and molecular data with physical 3D reconstructions, we provide a comprehensive framework for assessing post-traumatic bone damage and infection. Among the tested technologies, PolyJet and SLA printing technologies emerged as the most accurate methods for replicating complex bone architecture, underscoring their suitability for high-precision clinical and forensic applications. In contrast, FDM-based systems, although less precise, offer advantages in terms of cost efficiency, accessibility, and ease of material handling. Our updated analysis includes a detailed comparison of printer-specific advantages and limitations. PolyJet offers excellent surface fidelity and supports sterilisation. SLA combines high resolution with faster print times, and FDM technologies, although more variable, are widely available and scalable for prototyping purposes. Translating high-resolution imaging data into tangible 3D models offers new opportunities for personalised medicine, including surgical planning, implant fitting, and trauma simulation. Moreover, in forensic science, such models enhance anatomical visualisation, ballistic trajectory reconstruction, and legal documentation of injuries. As additive manufacturing technologies continue to evolve, their integration with advanced imaging and molecular diagnostics will drive the next generation of patient-specific interventions. Our findings emphasise the value of interdisciplinary innovation—linking digital imaging, materials science, and clinical application—to improve the precision, reproducibility, and translational impact of modern biomedical and forensic solutions.

## Figures and Tables

**Figure 1 biomedicines-13-01742-f001:**
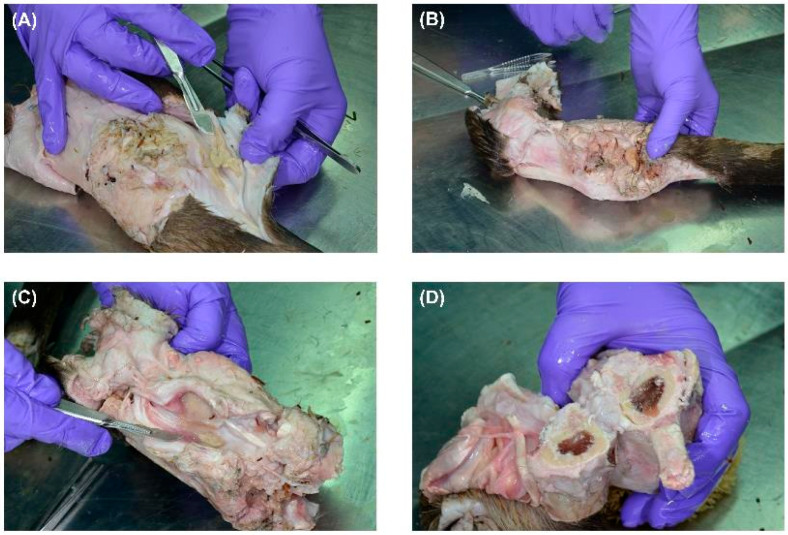
Stepwise dissection of a gunshot-injured limb showing extensive soft tissue destruction and infection. (**A**) Initial dissection reveals a gunshot entry wound with fibrin deposits and pus formation, suggesting secondary infection. The surrounding skin shows necrotic changes. (**B**) Deeper dissection exposes severe muscle fibre disruption along the bullet tract, with loss of structural integrity and cavitation effects. Visible pus pockets indicate a progressing infection. (**C**) Advanced dissection highlights deep tissue involvement, with necrotising changes, tissue liquefaction, and extensive purulent material consistent with infected gunshot trauma. (**D**) A cross-section of the bone shows localised damage from projectile impact, with fractures and bone marrow discolouration. The absence of purulent accumulation suggests early-stage bone involvement without established osteomyelitis.

**Figure 2 biomedicines-13-01742-f002:**
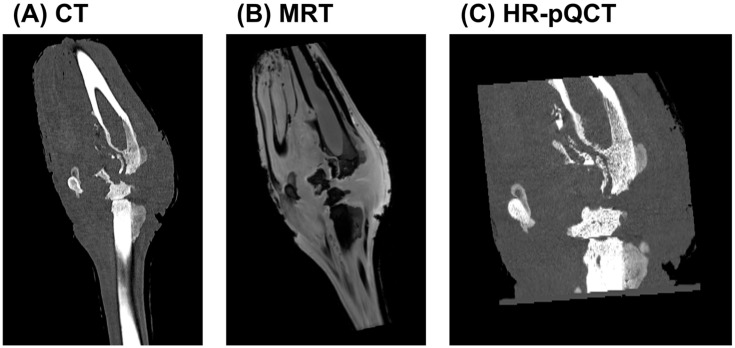
Multimodal imaging analysis of a gunshot-induced limb injury. (**A**) CT imaging demonstrates extensive bone comminution, cortical disruption, and periosteal reaction, suggestive of high-energy trauma and potential osteomyelitis. (**B**) MRI visualises soft tissue disruption, oedema, and altered bone marrow signal intensity, indicating haemorrhage, necrosis, and possible abscess formation. (**C**) HR-pQCT provides high-resolution visualisation of trabecular collapse, cortical erosion, and early bone sequestration, consistent with localised osteolytic changes and early-stage osteomyelitis.

**Figure 3 biomedicines-13-01742-f003:**
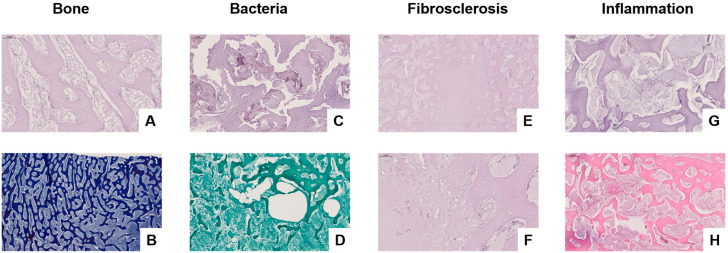
Histological assessment of infected bone and surrounding soft tissue using PAS, CAb, Grocott’s Methenamine Silver (GMS), and Haematoxylin and Eosin (H&E) staining. (**A**) PAS staining reveals intact trabecular bone structure with fatty marrow and no signs of inflammation (magnification level 20×). (**B**) CAb staining of preserved bone shows no evidence of fungal elements (magnification level 5×). (**C**) PAS staining highlights trabecular degradation and disrupted bone architecture, suggestive of microbial colonisation (magnification level 20×). (**D**) GMS staining identifies bacterial biofilm-like aggregates (“bacterial turf”) within porous bone, with no fungal hyphae present (magnification level 5×). (**E**,**F**) PAS staining of fibrotic soft tissue demonstrates extracellular matrix accumulation and fibroblast-rich remodelling, consistent with fibrosclerosis (magnification level 5× (**E**) and 20× (**F**)). (**G**) PAS staining of inflamed bone reveals structural disorganisation and mononuclear cell infiltration (magnification level 10×). (**H**) H&E staining confirms active inflammation with dense mixed-cell infiltrates, osteolysis, fat necrosis, and vascular congestion (magnification level 5×).

**Figure 4 biomedicines-13-01742-f004:**
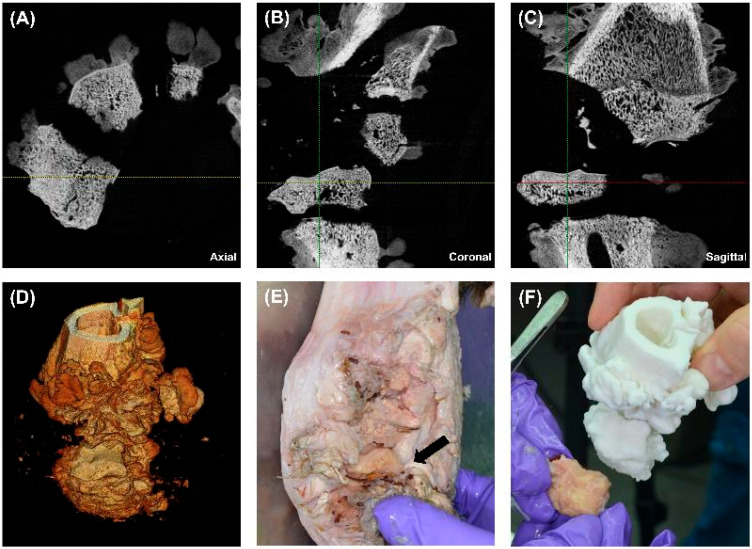
Multimodal imaging, volumetric reconstruction, and 3D printing for the assessment of gunshot-induced bone damage. (**A**) Axial HR-pQCT scan showing multiple displaced bone fragments, cortical breaches, and early osteolytic changes (zoom 2×). (**B**) Coronal HR-pQCT scan revealing vertical displacement of bone segments, cortical discontinuities, and trabecular disruption, indicating severe structural instability (zoom 2×). (**C**) Sagittal HR-pQCT scan displaying full-thickness cortical damage, trabecular disorganisation, and periosteal reaction suggests a chronic inflammatory response (zoom 2×). (**D**) 3D volume rendering of the affected bone, visualising the extent of bone destruction, sequestration, and structural irregularity. (**E**) Gross dissection of the affected limb, highlighting an exposed necrotic bone fragment (black arrow) embedded within inflamed and fibrotic soft tissue. (**F**) Comparison between the extracted bone fragment and a 3D-printed model, demonstrating the accuracy of CT-based digital reconstructions for forensic and surgical applications.

**Figure 5 biomedicines-13-01742-f005:**
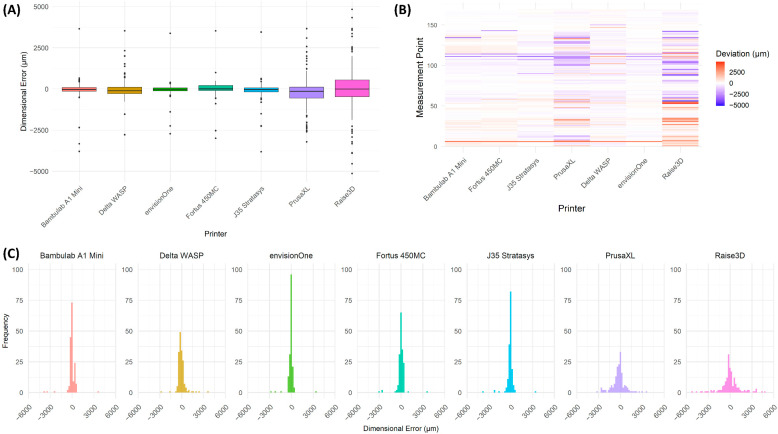
Statistical Analysis of Dimensional Deviations Across Different 3D Printers. (**A**) Box plot showing the distribution of deviations for each 3D printer, indicating variation in dimensional accuracy. The interquartile range (IQR) represents the spread of deviations, with individual outliers displayed as black dots. Printers such as Bambulab A1 Mini, envisionOne, and Fortus 450MC_02 exhibited lower deviation variability, whereas PrusaXL and Raise3D showed greater deviation spread, suggesting higher inconsistency in print accuracy. (**B**) Heatmap visualisation of deviation distribution across different measurement points, with colour intensity representing deviation magnitude. Red indicates positive deviation (over-extrusion or expansion), blue represents negative deviation (material shrinkage or under-extrusion), and white corresponds to minimal deviation. Printers such as Bambulab A1 Mini and envisionOne exhibited more uniform distributions, while PrusaXL and Raise3D displayed significant deviations across multiple regions, reflecting higher geometric inconsistency. (**C**) Histogram representation of deviation frequencies, illustrating the distribution of deviations for each printer. Printers with narrower peaks demonstrate higher accuracy and consistency, while broader distributions indicate more significant print variability.

**Figure 6 biomedicines-13-01742-f006:**
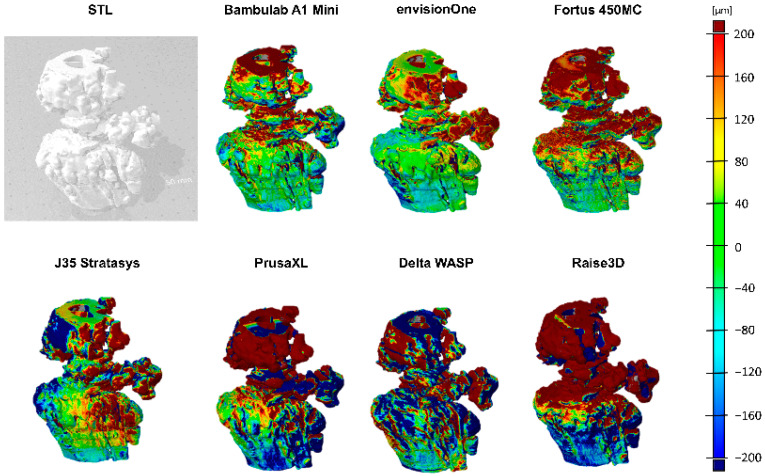
The 3D deviation mapping of 3D-printed models. The grayscale image (**top left**) shows the reference model. False-colour maps visualise surface deviations, where red indicates positive deviation, blue negative deviation, and green represents nominal accuracy. Variations across different views and colour distributions reveal differences in print precision, material behaviour, and process fidelity among tested 3D printing technologies.

**Table 1 biomedicines-13-01742-t001:** Three-dimensional printers and printing parameters.

3D Printing Technology	Printer Model	Material Used	Layer Thickness	Post-Processing
PolyJet	J35 (Stratasys)	VeroContactClear	18.75 µm	Waterjet cleaning
Fused Deposition Modeling (FDM)	Fortus 450MC (Stratasys)	ABS Thermoplastic	0.254 mm	Chemical bath support removal
	RAISE3D E2	PLA	0.2 mm	Water-soluble support
	Prusa XL	PLA + BVOH Support	0.2 mm	Support dissolved in water
	Bambu Lab A1 Mini	PLA	0.2 mm	Manual support removal
	WASP (Delta Printer)	PLA	Variable	Manual support removal
Stereolithography (SLA)	envisionOne	E-RigidForm Char	50 µm	UV post-curing and mechanical support removal

**Table 2 biomedicines-13-01742-t002:** Dimensional accuracy of different 3D printers based on surface deviation analysis. Mean deviation, standard deviation, minimum, and maximum deviations (in micrometres) were calculated from point cloud comparisons against a reference model. Negative values indicate material deficits (undersized features), while positive values represent excess material (oversized features). Fused Deposition Modelling (FDM). Stereolithography (SLA).

Printer Model	Technology	Mean Deviation (µm)	Std Dev (µm)	Minimum (µm)	Maximum (µm)
Bambulab A1 Mini	FDM	−15.62	569.41	−3793	3652
envisionOne	SLA	−10.31	452.16	−2897	3205
Fortus 450MC	FDM	12.06	510.85	−2998	3528
J35 Stratasys	PolyJet	−84.20	521.80	−3812	3456
PrusaXL	FDM	−209.77	971.15	−3211	3665
Delta WASP	FDM	−98.53	672.90	−4078	3985
Raise3D	FDM	−46.49	554.16	−2779	3525

**Table 3 biomedicines-13-01742-t003:** Benchmarking Parameters of 3D Printing Technologies.

Printer Model	Technology	Layer Thickness (µm)	Print Time (h)	Material Used (g)		Post-Processing Time (h)	Sterilisation Suitability
Bambulab A1 Mini	FDM	200	6.40	84	Pla Overture Matte Blue	0.5	Not specified
envisionOne	SLA	50	4.15	Not specified	E-RigidForm Char	Not specified	Not specified
Fortus 450MC	FDM	254	9.24	89	ABS-M30i	1–4	Gamma, EtO
J35 Stratasys	PolyJet	18.75	9.53	196	VeroContactClear	0.08	Gamma, Steam
PrusaXL	FDM	200	12.00	81	Pla Premium 1.75 mm Arctic White Polydissolve S1 water-soluble support	2	Not specified
Delta WASP	FDM	Variable	Not specified	Not specified	FILOALFA^®^ ABS	Not specified	Not specified
Raise3D	FDM	200	31.00	67.8	Pla Premium 1.75 mm Arctic White Polydissolve S1 water-soluble support	96 (water-soluble)	Not specified

## Data Availability

The data presented in this study are available upon request from the corresponding author.

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
