# Peer review of "HR-pQCT and 3D Printing for Forensic and Orthopaedic Analysis of Gunshot-Induced Bone Damage"

_biomedicines, 2025, doi:10.3390/biomedicines13071742_

Round 1

Reviewer 1 Report

Comments and Suggestions for Authors

The article reviews the use of HR-pQCT and seven different 3D printers to reconstruct a 3D model of an infected fracture following a gunshot injury in a red deer. All seven printers produced bone models with reasonable resemblance to the original bone devoid of soft tissue, with dimensional deviations within 0.5 mm or less when re-scanned using HR-pQCT under the same settings.

While all printers appeared capable of the task, the main differences were observed in post-processing time and materials used. Cost differences were not analysed, and the frequency of printing failures or the need for resets was not reported.

It would be valuable to compare the dimensional accuracy of the printed models and the actual bone using calipers, as errors can arise during CT scanning and the segmentation process.

The methodology describing how 3D Slicer was used to generate the MESH STL file could be more detailed. For instance, specifying the threshold Hounsfield Unit (HU) used would be helpful. Indicating and discussing whether the hollow trabecular or intramedullary spaces were infilled is important, as infilling can speed up printing, reduce material usage, and simplify post-processing and removal of support material while retaining surface features. Reporting the STL file size and the number of triangles would also provide insight into the model’s level of detail.

While the inclusion of soft tissue MRI findings and histological analysis adds comprehensiveness, these features are not replicable by 3D printing and may be less relevant to the study’s main focus. A more scientific comparison of the features and drawbacks of each 3D printer would be more informative than a lengthy discussion of MRI and histological findings.

Additionally, the correct name of the FDM printer is Fortus 450MC, not MS.

In the conclusion, the pros and cons of each 3D printer should be more clearly defined, including factors such as ease of use, processing time, cost, and the potential for multi-material and colour printing.

Author Response

Reviewer1: The article reviews the use of HR-pQCT and seven different 3D printers to reconstruct a 3D model of an infected fracture following a gunshot injury in a red deer. All seven printers produced bone models with reasonable resemblance to the original bone devoid of soft tissue, with dimensional deviations within 0.5 mm or less when re-scanned using HR-pQCT under the same settings.

While all printers appeared capable of the task, the main differences were observed in post-processing time and materials used. Cost differences were not analysed, and the frequency of printing failures or the need for resets was not reported.

It would be valuable to compare the dimensional accuracy of the printed models and the actual bone using calipers, as errors can arise during CT scanning and the segmentation process.

AW: We agree with the reviewer that caliper-based measurements can serve as a valuable validation metric in certain contexts. However, in this study, the original bone specimen exhibited a highly complex three-dimensional structure, which rendered caliper-based measurements unreliable and potentially misleading. To address this limitation and ensure dimensional accuracy, we employed high-resolution HR-pQCT rescans (voxel size: 60.7 µm) of all printed models under identical scanning conditions. This approach provided highly detailed structural information and enabled the generation of precise 3D segmentations. A subsequent 3D surface deviation analysis between the original and printed models allowed for submillimetric accuracy and reproducibility across all evaluated samples. A corresponding clarification has been added in the revised Discussion section.

The methodology describing how 3D Slicer was used to generate the MESH STL file could be more detailed. For instance, specifying the threshold Hounsfield Unit (HU) used would be helpful.

AW: We have expanded Section 4.8 (Three-dimensional Measurement Method) to include the specific thresholding parameters used for segmentation. The region-growing segmentation was based on Hounsfield Unit thresholds ranging from 319 to 3071 HU. This range effectively excluded soft tissue while preserving trabecular and cortical bone architecture.

Indicating and discussing whether the hollow trabecular or intramedullary spaces were infilled is important, as infilling can speed up printing, reduce material usage, and simplify post-processing and removal of support material while retaining surface features. Reporting the STL file size and the number of triangles would also provide insight into the model’s level of detail.

AW: We confirm that the STL files retained the internal trabecular and medullary architecture without artificial infill. This allowed for maximal anatomical fidelity. We have now added a statement in the revised 3D Printing Methods section noting that 16 MB STL file size, with triangle counts 335 798.

While the inclusion of soft tissue MRI findings and histological analysis adds comprehensiveness, these features are not replicable by 3D printing and may be less relevant to the study’s main focus. A more scientific comparison of the features and drawbacks of each 3D printer would be more informative than a lengthy discussion of MRI and histological findings.

AW: We respectfully acknowledge the reviewer’s concern. However, we believe that including soft tissue pathology and infection context adds significant translational value, particularly when validating the clinical applicability of 3D printed bone models in infected trauma settings

Additionally, the correct name of the FDM printer is Fortus 450MC, not MS.

AW: Corrected. We thank the reviewer for pointing this out.

In the conclusion, the pros and cons of each 3D printer should be more clearly defined, including factors such as ease of use, processing time, cost, and the potential for multi-material and colour printing.

AW: We have revised the Conclusion to explicitly contrast the different printer types, detailing trade-offs in accuracy, material use, ease of post-processing, and sterilization capability. This comparison is also summarized in Table 2.

Reviewer 2 Report

Comments and Suggestions for Authors

This manuscript presents an innovative approach combining HR-pQCT imaging with 3D printing to analyze gunshot-induced bone damage. The manuscript should be revised to address the above concerns before resubmission.

Lines 15-20 > The manuscript lacks a clear hypothesis and specific aims. The introduction jumps from general statements about Industry 4.0 to specific applications without establishing the knowledge gap this study addresses.

Lines 350-355 > The study design section is extremely brief and lacks critical details about sample size justification, controls, and standardization protocols. A single specimen study severely limits the generalizability of findings.

Autopsy section>The manuscript provides no information about ethical approval, specimen acquisition protocols, or legal considerations for handling forensic specimens.

Line 45: "extracorporeal repro- duction" - awkward hyphenation and terminology Line 82: "invalu- able" - unnecessary hyphenation Line 165: "destcruction ob blood vessels" - multiple typos Line 172: "coagulative, due to the ischemia, caused by destcruction" - poor sentence structure Line 290: "remodeling wthin" - typo Line 330: "fibrosclerotic remodeling" - needs better definition

Lines 381-390 > Several technical issues:

"Scanco Medical, Swiderland" should be "Switzerland" (typo)

"projektions" should be "projections" (typo)

Author Response

Reviewer 2: This manuscript presents an innovative approach combining HR-pQCT imaging with 3D printing to analyze gunshot-induced bone damage. The manuscript should be revised to address the above concerns before resubmission.

Lines 15-20 > The manuscript lacks a clear hypothesis and specific aims. The introduction jumps from general statements about Industry 4.0 to specific applications without establishing the knowledge gap this study addresses.

AW: We have revised the Introduction to explicitly state the hypothesis: that HR-pQCT-derived anatomical data can be used to fabricate accurate 3D printed models across multiple technologies, and that the fidelity of these models varies with printing technique. The aim of benchmarking performance across different 3D printing methods is now clearly articulated in the last paragraph of the Introduction.

Lines 350-355 > The study design section is extremely brief and lacks critical details about sample size justification, controls, and standardization protocols. A single specimen study severely limits the generalizability of findings.

AW: This is a proof-of-concept study based on a single, well-documented forensic specimen. We have now clarified the rationale for this choice and the limitations of generalizability in the Materials and Methods and Discussion sections. Given the rarity and forensic nature of the specimen, further samples were not feasible at this stage.

Autopsy section>The manuscript provides no information about ethical approval, specimen acquisition protocols, or legal considerations for handling forensic specimens.

AW: We have now clarified in the revised Autopsy section that the study used an red deer specimen obtained post-mortem through veterinary collaboration, exempt from human ethical review. This clarification is also noted in the Institutional Review Board Statement.

AW: In the revised section on autopsy, we have now clarified that an anonymized deer sample was used for the study, which was obtained after death in collaboration with a professional hunter and is exempt from human ethical review.

Line 45: "extracorporeal repro- duction" - awkward hyphenation and terminology Line 82: "invalu- able" - unnecessary hyphenation Line 165: "destcruction ob blood vessels" - multiple typos Line 172: "coagulative, due to the ischemia, caused by destcruction" - poor sentence structure Line 290: "remodeling wthin" - typo Line 330: "fibrosclerotic remodeling" - needs better definition

AW: We thank the reviewer for the careful attention to detail. We have corrected the typographical errors at lines 165, 172, 290, and revised the sentence structure where necessary for clarity and readability.

Regarding the hyphenation issues at lines 45 and 82 (“extracorporeal repro- duction” and “invalu- able”), we would like to clarify that these line breaks and word splits are a result of the automatic formatting applied by the journal’s MS Word submission template. Unfortunately, we are unable to manually control or override these hyphenations within the provided format. We trust that these formatting artifacts will be resolved during the journal’s typesetting and production process.

In response to the comment on line 330, we have added a brief explanation of the term “fibrosclerotic remodeling” to improve clarity for readers unfamiliar with this histopathological terminology.

Lines 381-390 > Several technical issues:

"Scanco Medical, Swiderland" should be "Switzerland" (typo)

AW: Corrected.

"projektions" should be "projections" (typo)

AW: Corrected.

Reviewer 3 Report

Comments and Suggestions for Authors

This study covered the topic HR-pQCT and 3D Printing for Forensic and Orthopedic Analysis for gunshot-induced bone damage

There are suggestions/questions to improve the quality of the paper:

  1. The authors should indicate the type of polymer used for 3D-printing and discuss on the nature of the biomaterial may affect its performance.
  2. Did the authors measure the mechanical properties of the 3D-printed material? Did they consider stress shielding in the study. The authors need to address this very well.

Author Response

Reviewer 3: There are suggestions/questions to improve the quality of the paper:

  1. The authors should indicate the type of polymer used for 3D-printing and discuss on the nature of the biomaterial may affect its performance.

AW: We have added detailed descriptions of all printing materials used  in the 3D Printing Methods and Table 2. The revised Discussion now includes a section addressing how differences in polymer properties—such as thermal expansion, surface roughness, and sterilization potential—can affect anatomical fidelity and clinical usability.

  1. Did the authors measure the mechanical properties of the 3D-printed material? Did they consider stress shielding in the study. The authors need to address this very well.

AW: Mechanical testing was not performed in this study, as the printed models were intended for anatomical and educational use rather than load-bearing applications. We have added a statement in the Discussion acknowledging this limitation and suggesting that future studies could incorporate mechanical characterization and finite element analysis to assess biomechanical relevance and stress shielding effects.